# Specifying Internet of Things Behaviors in Behavior-Driven Development: Concurrency Enhancement and Tool Support

**Bing-Yun Wang** 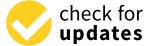, **Yi-Chun Yen** and **Yu Chin Cheng** *

Department of Computer Science and Information Engineering, National Taipei University of Technology, Taipei 10608, Taiwan
* Correspondence: yccheng@ntut.edu.tw

**Abstract:** The Internet of Things (IoT) systems are inherently distributed with many concurrent behaviors. In order to apply behavior-driven development (BDD), a proven agile practice of software development that brings many benefits, we must ensure that the specification of sequential and concurrent behaviors is supported at the specification level and that tool support is in place to execute the specification. This study proposes a minimal semantic enhancement to the Gherkin language, the most popular specification language in BDD, to distinguish sequential and concurrent behaviors. At the same time, a tool called `concurrentSpec` is developed to support the correct execution of specifications written in the enhanced Gherkin language. With two IoT examples involving both sequential and concurrent behaviors, it is shown that the enhanced Gherkin with `concurrentSpec` can correctly specify and execute the specifications, while the original Gherkin with existing tools is unable to do so. Hence, the contribution of this study is to eliminate a technical impediment for the IoT development community to adopt BDD and receive its benefits.

**Keywords:** Internet of Things; software engineering; concurrency; behavior-driven development

## 1. Introduction

The Internet of Things (IoT) development community has been embracing new software development methodologies to make IoT system development faster, better, and cheaper [1,2]. With success stories of the adoption of agile methodologies in IoT-related areas such as embedded systems and hardware [3–5], the IoT development community can be expected to continue to look into new methodologies, especially the ones with proven results in the broader software development community.

In this context, this paper explores applying *behavior-driven development* (BDD) [6] for IoT systems development. As a relatively new agile software development method evolved from *test-driven development* (TDD) [7], BDD broadens collaboration by moving from developer-centric tests to whole-team executable specifications. BDD has gained a lot of traction in the software development community. According to a recent vendor survey, 37%, 44%, and a projected 50% of software development teams were adopting BDD in 2017, 2018, and the 3–5 years beyond, respectively [8]. With an ample amount of the literature (e.g., [9–12]) and many supporting tools available (e.g., [13]), BDD makes a good candidate methodology for IoT system development. Indeed, we are beginning to see positive results in applying BDD for developing real-time embedded systems [14], automotive systems [15,16], and avionic systems [17]. In our own experience, we have also benefited from BDD in the development of a smart cone system for protecting road maintenance crews working on highways [18].

Key to the success of BDD is the patterns that encourage the development team to *specify collaboratively* and *illustrate with examples* [9–12], thereby minimizing the gaps of misunderstanding between members of the IoT development team including domain experts, analysts, hardware and software developers, and testers. The captured specifications are

recorded in textual descriptions written in a simple natural language template; an example is the *Given-When-Then* template of the Gherkin language [19]. Since well-written Gherkin specifications are readable and understandable by all team members, the specifications can be continuously refined throughout the life cycle of an IoT system for feature changes, extensions, and bug fixes. Further, through tool support, the Gherkin specifications of an IoT system are *executable specifications* that serve as *acceptance tests* to drive the subsequent coding and testing activities [9]. When frequently executed and maintained, the specifications turn into *living documentation* to replace the traditional requirements and specifications documentation that is prone to be out of sync with the current system behaviors [9,12].

Despite the merits of BDD described above, in developing the smart cone system which encompasses both sequential and concurrent behaviors [18], we found that the Gherkin language does not yet support the specification of concurrent behaviors. This inadequacy created some difficulties in writing specifications with Gherkin for our interdisciplinary team consisting of traffic safety specialists, hardware engineers, embedded system developers, and software developers. In particular, the traffic safety specialists were unable to use Gherkin to precisely specify the concurrent behaviors of the smart cone system and had to depend on the developers to interpret the intended concurrent behaviors at the implementation level. We will further show that the concurrency inadequacy of Gherkin can lead to specification errors with two IoT examples in Section 3.

To fix this inadequacy, we propose an enhancement to Gherkin to support the specification of concurrent behaviors. We follow the *parsimony principle* [20] by not adding new keywords to Gherkin. Instead, we introduce the concept of *sequential groups of concurrent steps* by limiting the use of Gherkin keywords `And` and `But` for concurrency. Specifically, the groups are interpreted sequentially in the order they appear in a Gherkin specification. Each group is led off by a step that begins with keyword `Given`, `When`, or `Then`. Inside each group, the steps beginning with keyword `And` or `But` are interpreted as steps concurrent with the leading step. The enhancement brings concurrency support without adding lexical complexity of Gherkin. Existing sequential Gherkin specifications can be translated into the enhanced Gherkin by simply replacing `And` and `But` keywords in the steps with the closest `Given`, `When`, or `Then` keyword before them, respectively.

To fully support the concurrency enhancement, we develop an embedded Gherkin tool support named `concurrentSpec` in Python [21]. In addition, `concurrentSpec` also supports the continuation of execution after a step fails within a `Then` group. It is shown that with the proposed Gherkin enhancement and `concurrentSpec`, the specification errors found in the two motivating IoT examples can be fully mitigated. While IoT is our main focus in this paper, as a side note, the proposed concurrency enhancement and tool support can also be used to develop ordinary distributed systems with BDD.

The rest of this paper is organized as follows. Section 2 covers the background of BDD and tool supports. Section 3 presents the two examples demonstrating the specification errors that could result from using the current Gherkin language and tool support for specifying concurrent behaviors of IoT systems. Section 4 presents the proposed concurrency enhancement to the Gherkin language and the supporting tool `concurrentSpec`. The resolution of the problems of the motivating examples are shown through simulations in Section 5. Section 6 discusses the consequences of the enhancement and related issues. Finally, Section 7 summarizes the contribution of this work.

## 2. Background

Since BDD derives from TDD, we begin by describing TDD. TDD is a widely adopted software development practice that advocates writing automated tests before writing production code [7,22]. Incrementally and iteratively, a test is added that specifies and verifies the functionality of a specific aspect of a small individual component. Then, just enough production code is written to pass the added test as well as all tests that were previously passing. The code is refactored for a higher quality before ending an iteration.

TDD has been applied at all levels from acceptance testing to unit testing [23,24]. TDD has demonstrated successes among early adopters in IoT development (e.g., [4]).

As a derivative of TDD, BDD further aims to bridge the communication gap between domain experts, business analysts, developers, and testers by using a natural language in a simple but special format to specify software behaviors through concrete examples [6,25,26]. While different formats are available [13], this paper focuses on Gherkin [19], a popular language where specifications are written in plain text and supported by many tools.

### 2.1. Syntax and Semantics of Gherkin

The class diagram in Figure 1 shows the syntactic structure of keywords of Gherkin. At the top level is the keyword *feature*, which represents a unit of functionality that the software under development will have upon completion [27], such as transferring money between two accounts in a banking application, booking a room in a hotel application, and so on. A feature contains a number of *scenarios*, where a scenario is a concrete example that describes how the feature plays out under the specific circumstance. For example, the behaviors of transferring money will be different, respectively, depending on the source account having a sufficient or an insufficient balance before a transfer.

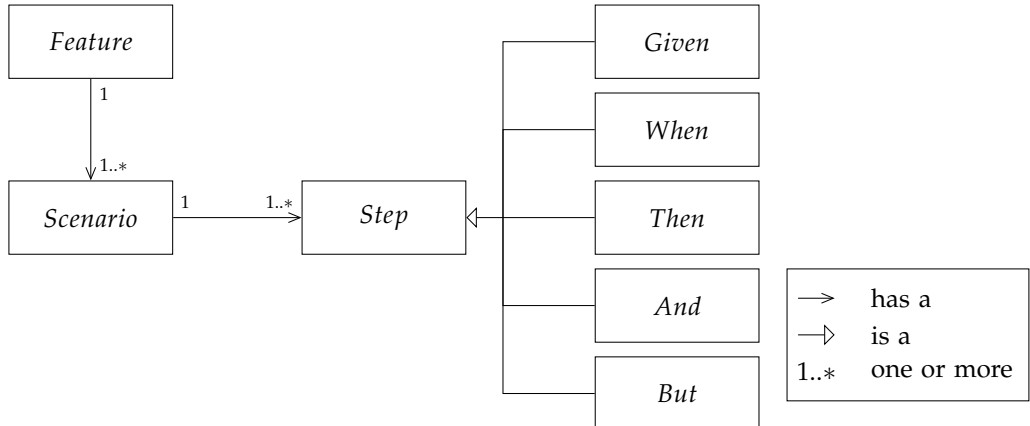

**Figure 1.** The structure of Gherkin keywords in a feature.

A scenario contains several *steps* which are interpreted *sequentially in the order of appearance*. A step begins with one of the keywords *Given*, *When*, or *Then*:

- A *Given* step gives the precondition or the initial context of the scenario;
- A *When* step gives the event or action that sets the scenario in motion;
- A *Then* step gives the post-condition reflecting the effect of the event or action.

If a scenario contains several successive steps all beginning with the same keyword, each of the subsequent keywords can be replaced by an *And* or a *But* for better readability without changing its meaning; see Section 3 for an example. A feature and its constituent scenarios are stored in a plain text file with the `.feature` extension.

### 2.2. Tool Support

A feature is an *executable specification* through tool support. The execution either passes or fails, and a feature is completely implemented when all of its scenarios pass without failure. Through a proper BDD tool support, such as `Cucumber` for Java, JavaScript, and Ruby [28], `behave` for Python [29], `SpecFlow` for C# [30], and so on, each step found in a feature file is translated into a *step definition*, a function or an object method in the target programming language. Developers then work on the step definitions, presumably in the TDD style, to complete the design and implementation of the feature.

Despite the multitude of target programming languages supported, each of the tools in the Gherkin family achieves executable specification in a way similar to what is described next.

Initially, when given a feature file as its input, a BDD tool generates a function (or an instance method) in the target language as the step definition for each different step inside the feature file. In all the aforementioned BDD tools, the tool-generated code for a step definition function prints a message to show that the step is not implemented yet. Developers and testers replace the generated code with: (1) code that calls the functions and object methods to be implemented for the feature; and (2) code that checks their correctness.

Assume that the step definition functions of a scenario are implemented. When tool is run again with the feature file containing the scenario as input, the steps in the scenarios are matched with its step definition functions, which are executed sequentially in the order of appearance. All existing tools adopt the sequential interpretation.

### 2.3. Failing-Fast

When a step in a scenario fails in execution, the scenario fails and terminates immediately, skipping the subsequent steps. This is referred to as *failing-fast* [31] and is the default behavior for all tools supporting Gherkin. The default behavior is sometimes inconvenient, (for example, in checking multiple independent post-conditions in separate `Then` steps following a `When` step,) and many tools now provide the option to turn off the failing-fast feature to continue executing the remaining steps.

It is interesting to note that support for turning off failing-fast varies by tool. For example, `behave` allows turning off the failing-fast feature for the whole scenario [32]. Once the option is turned off, every step in the scenario will be executed regardless of the result of previous steps. In comparison, although `Cucumber` does not support turning off the failing-fast feature [33], an open-source extension for `Cucumber-JVM` [34] allows turning off the failing-fast feature at the individual steps.

As will be shown in the two motivating examples in Section 3, the sequential interpretation of the steps in a scenario makes it difficult and sometimes impossible to specify concurrent behaviors in IoT and distributed systems. Further, in specifications that involve both concurrent and sequential behaviors, the description of the behaviors cannot be easily supported by simply turning off the failing-fast feature.

## 3. Two Motivating Examples

We present two motivating examples to argue for the need of supporting the specification of concurrent behaviors. Further, the supporting tool must properly allow turning off the failing-fast feature at the step level in the context of concurrency. The two examples, *scheduled sprinkling* (Section 3.1) and *lift safety in emergency* (Section 3.2), are kept simple but sufficient to reveal the problems the two core issues can cause.

### 3.1. Example: Scheduled Sprinkling

*A lawn sprinkler system for home use is needed. Three sprinklers, each with a head connected by a pipe to a common water supply, are installed. Water emits from the sprinkler heads according to a schedule set by the owner through a controller. When it is time to sprinkle water, each sprinkler head begins emitting water within 5 s.*

A plausible specification for capturing this requirement is by the Gherkin scenario of Figure 2. In this scenario, the three sprinklers have been set up (line 2), and the scheduled time has been set to 4:00:00 a.m. (line 3). When the scheduled time arrives (line 4), all three sprinklers should emit water within 5 s (lines 5, 6, and 7).

```
1    Scenario: scheduled sprinkling
2       Given three sprinklers A, B, and C
3         And the scheduled time has been set to 4:00:00 am
4        When the time is 4:00:00 am
5        Then sprinkler A should emit water within 5 seconds
6         And sprinkler B should emit water within 5 seconds
7         And sprinkler C should emit water within 5 seconds
```

**Figure 2.** Scheduled sprinkling.

It should be clear from the problem statement that the three sprinklers should act *concurrently*; a failure in any of the three should not prevent the other two from exhibiting the required behavior. For instance, assume that the pipes branching off to sprinklers A and C, respectively, are clogged, but the pipe to sprinkler B is normal. When the scenario is executed, sprinkler A and sprinkler C should fail to emit water within 5 s (line 5 and line 7), whereas sprinkler B should emit water normally (line 6).

Under the current policy of the Gherkin language to interpret the steps sequentially and the policy of the existing supporting tools and to fail fast, the failure in the Then step for checking sprinkler A terminates the scenario, preventing sprinklers B and C from being checked. The result is that the developers will only know that the scenario has failed due to sprinkler A but would not know the status of sprinklers B and C. The developers will only know that sprinkler B would act normally and sprinkler C would fail by re-running the scenario after fixing the clog in sprinkler A. This is clearly inefficient.

It can be noted that turning off the failing-fast feature does not solve the problem either so long as the steps are interpreted sequentially. After sprinkler A fails, five seconds have already expired, causing the normal-functioning sprinkler B to fail. More of the anomaly introduced by the sequential interpretation of steps by Gherkin and the failing-fast feature of tool support will be discussed in Section 5.

On the other hand, the result of executing the scenario would reveal failures of sprinklers A and C and success of sprinkler B if the three assertions are executed *concurrently*. Further, the result remains so regardless of the failing-fast feature of the tool used.

*3.2. Example: Lift Safety in Emergency*

The next example is adapted from the problem descriptions in [35,36]:

*A lift that guarantees safe operation is needed. Normally, the lift provides timely transport service to passengers by taking requests and stopping at the requested floors. However, in the case of an emergency, such as a power outage, a mechanical/electrical malfunction, or an earthquake, the lift must come to a full stop at the nearest floor in the direction of travel, open its doors and keep them open, give safety warnings across all floors, and cancel any outstanding requests.*

In Figure 3, two scenarios are used to specify the requirements. The first scenario (lines 1–7) stipulates that when an emergency has been detected (line 3), the first Then step requires for the lift to stop (line 4), give warning (line 5), and cancel outstanding requests (line 6). The next Then step (line 7) requires that the doors should be open after a short delay for the passengers to exit safely.

The first scenario specifies the "normal path" of the lift system's response to an emergency. It should be kept in mind that it is possible for any step to fail. In the example, the lift can fail to stop at the first Then step (line 4). Under *the failing-fast policy*, the warning step (line 5), the cancellation step (line 6), and the subsequent Then step (line 7) are skipped. Omission for the former two steps can cause confusion for passengers, but omission of the latter step can pose great risk for passengers. If the doors fail to remain closed, a large gap can open up between the lift and the landing sill, which could put the passengers inside the lift in a life-threatening danger as they may attempt to get out of the lift through the gap. The safety requirement must steer the lift and doors clear of such a situation [37].

```
 1  Scenario:  emergency  braking  and  warning  over  normal  requests
 2      Given  an  outstanding  request  for  lift  to  visit  a  floor
 3       When  an  emergency  has  been  detected
 4       Then  lift  is  stopped  at  nearest  floor  in  direction  of  travel
 5        And  emergency  indicator  should  be  turned  on
 6        And  request  should  be  canceled
 7       Then  lift  doors  should  be  open  within  5  seconds
 8
 9  Scenario:  ensuring  passenger  safety  if  the  lift  fails  to  stop
10      Given  an  emergency  has  been  detected
11        But  lift  has  not  stopped  properly  at  a  floor
12       Then  lift  doors  should  remain  closed
```

**Figure 3.** Lift operations: safety over serving normal requests in case of an emergency.

It is not possible to fix the problem by turning off the failing-fast feature of the supporting tool used to run the scenario:

- When the feature is turned off for the whole scenario, the two And steps (lines 5–6) and the Then step (line 7) will be executed. The execution of the two And steps are desirable and correctly checks the warning and cancellation of outstanding requests. Unfortunately, the checking of the lift doors of the Then step can be *misleading*: if it passes, the doors are open when the lift is either in motion or does not stop aligning with the landing, putting the passengers in danger; and if it fails, the doors are correctly closed for passenger safety;
- When the feature can be turned off at the step level, failing-fast at the first Then step (line 4) must be turned off to guarantee execution of the warning and cancellation steps, but doing so could lead to the execution of checking lift doors and obtain a misleading result when the lift has failed to stop. On the other hand, keeping the failing-fast behavior prevents warning and cancellation steps if the first Then step fails. Either way, potential safety checking omissions exist.

In summary, the checking of the lift doors of the Then step should take place *only if* the lift has successfully stopped at the first Then step (line 4), and the checking of warning and request cancellation should be executed *only if* the lift has failed to stop.

For the safety requirement to be completely specified, the second scenario (lines 9–12) is necessary. If the lift fails to stop properly at a floor in an emergency (lines 10–11), the lift doors must remain closed to keep the passengers safe (line 12), presumably to wait for rescue. Note how the situation is encoded in the Given step and the But step.

Table 1 lists the combinations of the states of the lift and the lift doors. The lift may or may not be stopped at the nearest floor, and the lift doors may be open or closed. The first scenario in Figure 3 (line 7) checks that the doors are open in success or closed in failure when the lift is stopped at the nearest floor. The second scenario (line 12) checks that the doors are closed in success or open in failure when the lift is not stopped at a floor. As can be seen in Table 1, all combinations of states of the lift and the doors are covered, and the safety requirement is fully specified by the two scenarios of Figure 3.

**Table 1.** All combinations of states of the lift and the doors in checking safety.

| Lift | Lift Doors | Covered by Which Result in Figure 3 |
|---|---|---|
| stopped | open | passing at line 7 |
| stopped | closed | failing at line 7 |
| not stopped | open | failing at line 12 |
| not stopped | closed | passing at line 12 |

## 4. Concurrency Enhancement to Gherkin and Tool Support

We now propose a semantic enhancement of Gherkin to support specification of concurrent behaviors. In so doing, we adhere to the parsimony principle [20] to keep the enhancement minimal by not adding new keywords (Section 4.1). We also implemented a tool called `concurrentSpec` to support the enhancement and provide the step-level option to continue executing an enhanced Gherkin specification after a step fails (Section 4.2).

### 4.1. The Proposed Enhancement and Examples Resolved

Figure 4 shows the proposed enhancement in a class diagram. A feature contains one or more scenarios. A scenario is composed of one or more *sequential groups* that are executed in the order of appearance. Each sequential group has exactly one *lead step*, which can be a `Given` step, a `When` step, or a `Then` step. The lead step is followed by zero or more *concurrent steps* that take place simultaneously with the lead step. A concurrent step can be an `And` step or a `But` step. *A sequential group passes if and only if all of its constituent steps pass; it fails if any of its constituent step fails.* For brevity of reference, a sequential group beginning with a `Then` step is called a `Then` group, etc.

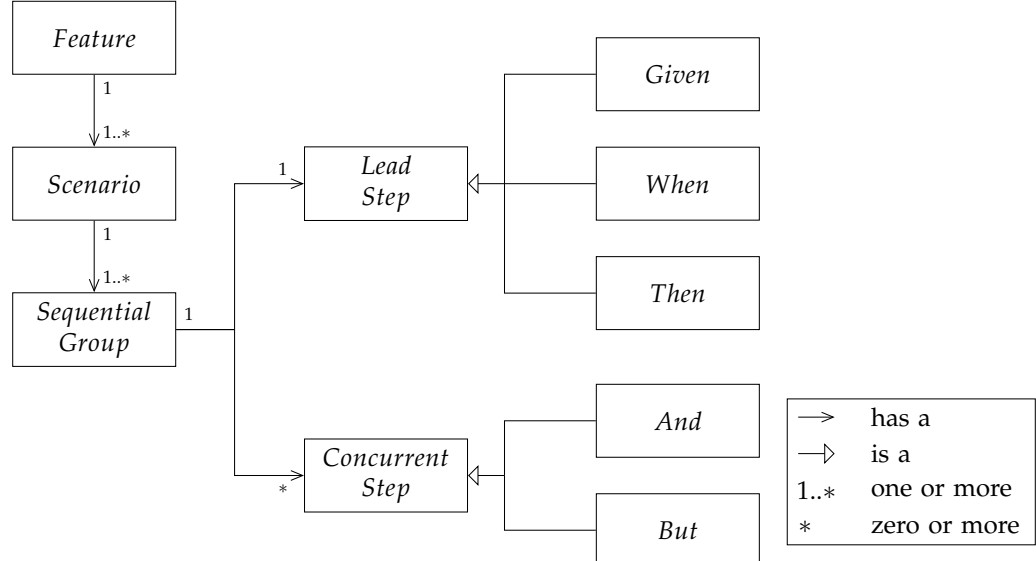

**Figure 4.** The proposed enhancement to Gherkin for specifying both sequential and concurrent behaviors.

With the enhancement, the scheduled sprinkling scenario of Figure 2 can now be interpreted correctly. The three steps of the `Then` group (lines 5–7) are executed concurrently. Thus, clogs in sprinkler A and sprinkler C are correctly detected with line 5 and line 7 failing, and sprinkler B passes the check at line 6.

In the lift safety scenario of Figure 3, the three steps in the `Then` group are executed concurrently (lines 4–6). Thus, even if the lift fails to stop (line 4), the checks of warning and request cancellation are still performed as required. More importantly, the check for doors opening in the `Then` step (line 7) is performed *only if* the lift indeed stops (line 4).

Note that a failure at line 5 and line 6 causes the `Then` group to fail, consequently preventing the check at line 7 even if line 4 passes. A remedy to this problem calls for the supporting tool to be capable of continuing execution in case either one of the concurrent steps at line 5 and line 6 fails; see Section 4.2.

### 4.2. Tool Support

The tool support of the proposed concurrency enhancement comes as an embedded *domain specific language* (DSL) [38]. It is named `concurrentSpec` and is implemented in Python. This subsection focuses on two features of `concurrentSpec`: (1) building and

executing the proposed sequential groups of concurrent steps and (2) continuing execution after failure at the step level.

Figure 5 shows the specification in `concurrentSpec` for the scheduled sprinkling scenario in Figure 2. The specification is written with code in Python rather than plain text. Since the scheduled time is set after the sprinklers are installed, the step at line 3 (Figure 5) replaces the `And` keyword (line 3 in Figure 2) with the `Given` keyword for the sequential execution. When executed, the program creates a data structure for organizing the steps as a sequence of groups of concurrent steps. In the meantime, it generates the skeleton code for step definitions of the scenario if they do not already exist; see Figure 6. Note that the skeleton code is later replaced with the actual step definition code written by developers and testers. The skeleton code for all steps simply fails by raising a `NotImplementedError` exception.

```
1  Scenario("scheduled sprinkling")\
2  .Given("three sprinklers A, B, and C")\
3  .Given("the scheduled time has been set to 4:00:00 am")\
4  .When("the time is 4:00:00 am")\
5  .Then("sprinkler A should emit water within 5 seconds")\
6  .And("sprinkler B should emit water within 5 seconds")\
7  .And("sprinkler C should emit water within 5 seconds")\
8  .execute()
```

**Figure 5.** Scenario of scheduled sprinkling in `concurrentSpec`.

```
1   class ScheduledSprinkling:
2       def __init__(self):
3           pass
4       def given_three_sprinklers_a_b_and_c(self):
5           raise NotImplementedError("given_three_sprinklers_a_b_and_c")
6       def given_the_scheduled_time_has_been_set_to_4_00_00_am(self):
7           raise NotImplementedError("given_timer_is_set_to_4_00_00_am")
8       def when_the_time_is_4_00_00_am(self):
9           raise NotImplementedError("when_the_time_is_4_00_00_am")
10      def then_sprinkler_a_should_emit_water_within_5_seconds(self):
11          raise NotImplementedError("
                then_sprinkler_a_should_emit_water_within_5_seconds")
12      def then_sprinkler_b_should_emit_water_within_5_seconds(self):
13          raise NotImplementedError("
                then_sprinkler_b_should_emit_water_within_5_seconds")
14      def then_sprinkler_c_should_emit_water_within_5_seconds(self):
15          raise NotImplementedError("
                then_sprinkler_c_should_emit_water_within_5_seconds")
```

**Figure 6.** Default step definition of the scheduled sprinkling scenario generated in `scheduled_sprinkling.py`.

### 4.2.1. Executing a Scenario

Figure 7 presents the interactions taking place after calling the `execute` method in a sequence diagram. Beginning with the call to the `execute` method of the scenario object (❶ in Figure 7), the `scenario` object creates an instance of `ScheduledSprinkling` of Figure 6 (❷). Next, the sequential groups in the data structure are executed in turn by calling the `run_all_steps` method (❸). Within a sequential group, each `step` object occupies a thread (❹–❻) and executes concurrently (❼). A thread calls the corresponding step definition function of the `ScheduledSprinkling` instance (❽–❿). Any error raised is caught by the sequential group (⓫). The execution of a sequential group ends when all steps in a sequential group finish execution (⓬). If any error is returned from the sequential group, the scenario raises `RuntimeError` which contains the information of all captured errors and terminates the execution (⓭); otherwise, it will return success (⓮).

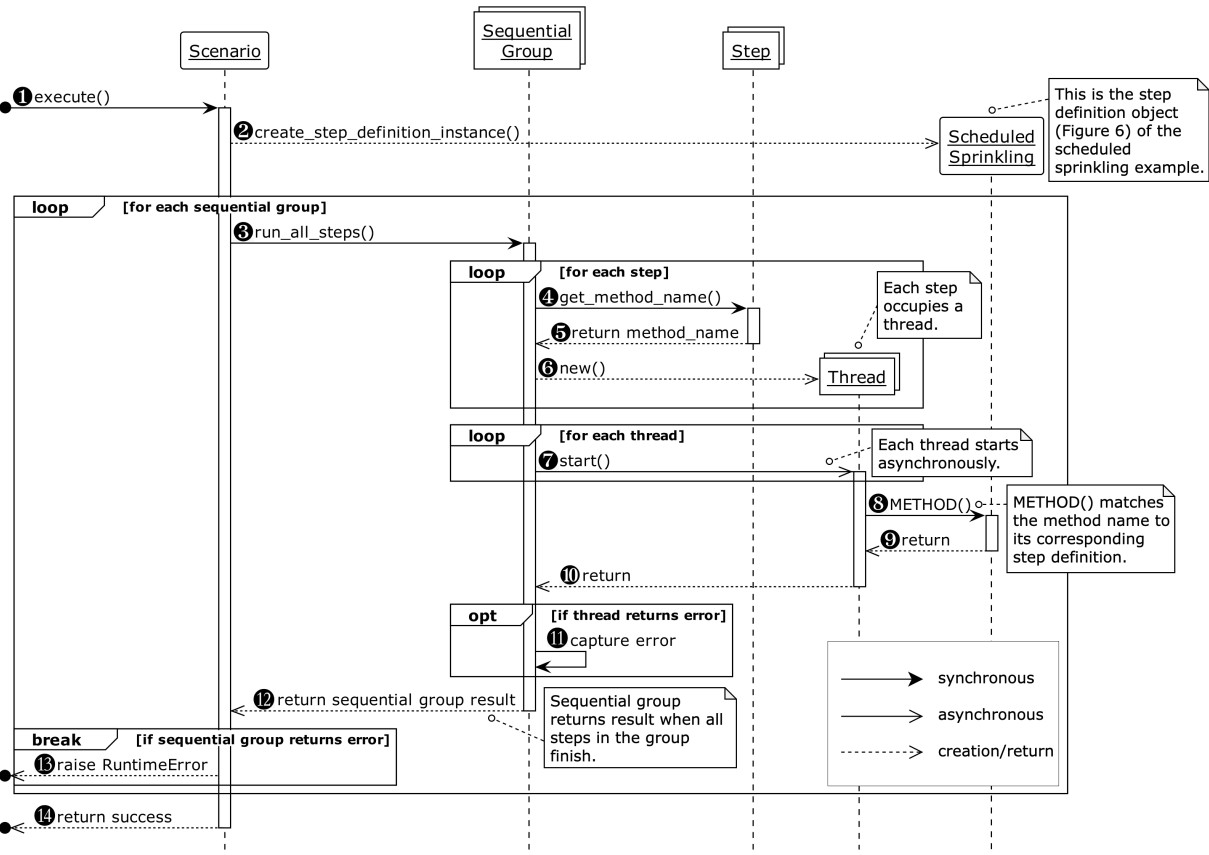

**Figure 7.** Sequence diagram of calling the `execute` method of the scheduled sprinkling scenario in `concurrentSpec`.

To support the passing/failing semantics of a `Then` group, an assertion failure thrown by a step in the group is caught and stored by `concurrentSpec`. At the conclusion of executing the group, a `RuntimeError` exception is raised to show information of all failing concurrent steps. Figure 8 shows the screen dump of executing the scenario in Figure 5 with both sprinklers A and C clogged.

```
E
==================================================================
ERROR: test_scheduled_sprinkling (__main__.TestScheduledSprinkling)
------------------------------------------------------------------
Traceback (most recent call last):
    ...
RuntimeError: Error(s) in the group:

    AssertionError from step: sprinkler A should emit water ...,
    error message: sprinkler A timeout

    AssertionError from step: sprinkler C should emit water ...,
    error message: sprinkler C timeout

------------------------------------------------------------------

Ran 1 test in 5.063s

FAILED (errors=1)
```

**Figure 8.** Screen dump of the scheduled sprinkling scenario with both sprinklers A and C clogged.

4.2.2. Continuing Execution after a Step Fails

As pointed out in the lift example in Section 4.1, in addition to the proposed concurrency enhancement to Gherkin, a full resolution needs the supporting tool to turn off the failing-fast, which means to continue executing the remaining steps after the warning step or the cancellation step fails. Since continuation after failure is a feature that receives much debate [39], in adding this feature to `concurrentSpec`, we have decided to restrict its applicable scope to steps in a `Then` group. Beyond keeping `concurrentSpec` simple to use, the rationale is as follows. In the case of the `Given` group, the precondition is established and possibly checked with an assertion. If the assertion fails, the precondition is not met, and it is pointless to continue execution since any further result is untrustworthy [11]. In the case of the `When` group, an event/action either takes place or not atomically, and an assertion is not needed. In both cases, it is best to retain the failing-fast policy to stop execution and report the errors immediately.

The continuation feature takes the form of an additional keyword argument to a step, *continue_after_failure*, which defaults to `false` and stops the execution upon an assertion failure. When set to `true`, execution continues even when the step fails. Since we restrict the use of this keyword argument to the steps in a `Then` group, setting *continue_after_failure = true* in a step inside a `Then` group simply means "when considering continuation for the `Then` group, ignore whether this step passes or fails". Figure 9 shows the correct specification by setting the keyword argument *continue_after_failure = true* for the two `And` steps (lines 5–6 in Figure 3). Thus, even if either one or both of them fail, the checking of doors opening at line 7 is executed when the lift stops successfully at line 4. On the other hand, when line 4 fails, since *continue_after_failure = false* by default, line 7 will not be checked, and any misleading result is avoided. Thus, Figure 9 fully resolves safety checking for the lift in the case of an emergency.

```
1  Scenario("emergency braking and warning over normal requests")\
2  .Given("an outstanding request for lift to visit a floor")\
3  .When("an emergency has been detected")\
4  .Then("lift is stopped at nearest floor in direction of travel")\
5  .And("emergency indicator should be turned on",
       continue_after_failure=True)\
6  .And("request should be canceled", continue_after_failure=True)\
7  .Then("lift doors should be open within 5 seconds")\
8  .execute()
```

**Figure 9.** Scenario of emergency braking and warning over normal requests in `concurrentSpec`.

## 5. Simulation Results

Through simulation, this section shows that both the concurrency enhancement of Gherkin and a tool support with the feature to continue execution after failure at the step level are necessary to enable correct specification and execution of concurrent behaviors in IoT systems. For comparison purposes, the simulations are also conducted with `behave` that does not support concurrency and with limited support for continuation after failure.

### 5.1. Scheduled Sprinkling

The simulations of scheduled sprinkling of Figure 2 demonstrate the necessity to have concurrency support. Since a sprinkler head is supposed to emit water within 5 s after the scheduled time arrives, a delay of time in the sprinkler is used to simulate clogging: no clogging if the delay is well below 5 s and clogging otherwise. Recall that all three sprinklers are supposed to act independently and simultaneously. Thus, our objective is to observe if clogging in any sprinkler incorrectly affects the checking of the behaviors of the remaining normal sprinklers.

The results are shown in Tables 2 and 3 with `behave` and `concurrentSpec`, respectively. Both tables are divided into four main columns. The first column shows the simulation number. The second column shows the time delays set for sprinklers A, B, and C, respectively. The third column presents the result by the three-tuple $(R_p, R_f, R_s)$, where $R_p$, $R_f$,

and $R_s$ is the number of times the assertion *passes*, *fails*, or is *skipped*, respectively. The last column indicates whether the result is correct (marked by "✓") or incorrect (marked by "✗").

**Table 2.** Simulation results of executing the scheduled sprinkling specification in Figure 2 with `behave`, where $R_p$, $R_f$, and $R_s$ is the number of times the assertion *passes*, *fails*, or is *skipped*, respectively. Each simulation is repeated 100 times.

| Simulation | Delay (Seconds) | | | $(R_p, R_f, R_s)$ with `behave` | | | Result Correct? |
|---|---|---|---|---|---|---|---|
| | A | B | C | A | B | C | |
| 1 | 0.2 | 0.5 | 0.3 | (100, 0, 0) | (100, 0, 0) | (100, 0, 0) | ✓ |
| 2 | 5.05 | 0.5 | 0.3 | (0, 100, 0) | (0, 0, 100) | (0, 0, 100) | ✗ |
| 3 | 4.99 | 0.5 | 0.3 | (10, 90, 0) | (9, 1, 90) | (8, 1, 91) | ✗ |

**Table 3.** Simulation results of executing the scheduled sprinkling specification in Figure 2 with `concurrentSpec`, where $R_p$, $R_f$, and $R_s$ is the number of times the assertion *passes*, *fails*, or is *skipped*, respectively. Each simulation is repeated 100 times.

| Simulation | Delay (Seconds) | | | $(R_p, R_f, R_s)$ with `concurrentSpec` | | | Result Correct? |
|---|---|---|---|---|---|---|---|
| | A | B | C | A | B | C | |
| 1 | 0.2 | 0.5 | 0.3 | (100, 0, 0) | (100, 0, 0) | (100, 0, 0) | ✓ |
| 2 | 5.05 | 0.5 | 0.3 | (0, 100, 0) | (100, 0, 0) | (100, 0, 0) | ✓ |
| 3 | 4.99 | 0.5 | 0.3 | (50, 50, 0) | (100, 0, 0) | (100, 0, 0) | ✓ |

In each table, three sets of simulations are conducted, each of which is repeated 100 times. Simulation 1 shows the results for the normal case using delays that are well under 5 s for all three sprinklers. Simulation 2 simulates clogging of sprinkler A by setting its delay to 5.05 s, while the remaining sprinklers are normal. Simulation 3 simulates the edge case of near-clogging of sprinkler A by setting its delay to 4.99 s, which is just under but very close to 5 s; the remaining sprinklers are normal.

Table 2 shows the simulation results for `behave`. In Simulation 1, all three sprinklers emit water within 5 s without failing, and none gets skipped. All have the correct outcome of $(R_p, R_f, R_s) = (100, 0, 0)$. In Simulation 2, the outcome for sprinkler A is correct at $(0, 100, 0)$ as it fails in all 100 repetitions. However, the outcomes for the independent sprinklers B and C at $(0, 0, 100)$ show that they are incorrectly skipped. In Simulation 3, sprinkler A exhibits intermittent behavior by passing 10 times while failing 90 times. The checks for sprinklers B and C are mostly skipped incorrectly due to the time used up by sprinkler A; when they are executed, they also incorrectly exhibit intermittent behaviors.

Table 3 shows the simulation results for `concurrentSpec` by executing the specification of Figure 5. The outcome of $(100, 0, 0)$ is also correct in all three sprinklers in Simulation 1. In Simulation 2, all three sprinklers give the correct result. Specifically, since the checks for all sprinklers are executed concurrently, clogging of sprinkler A has no effect on the remaining sprinklers. In Simulation 3, sprinkler A is intermittent with the outcome of $(50, 50, 0)$, but the remaining normal sprinklers both check out correctly by passing. (The better numbers in passing and failing in the outcome for sprinkler A are purely due to the implementation of `concurrentSpec`. The result can still only be interpreted as intermittent and is no better than that with `behave`.)

### 5.2. Lift Safety over Service in Emergency

Simulations for the lift safety example demonstrate that concurrency enhancement alone is not enough; the supporting tool must also provide the feature to continue executing the specification after a failure at the step level.

The simulations are limited to the first lift scenario in Figure 3. We assume that the `Given` step (line 2) and the `When` step (line 3) are executed successfully. Thus, the `Then` step

(line 4) is always correctly executed. The objective is to observe whether the two And steps (lines 5–6) and the last Then step (line 7) are incorrectly *executed* or *skipped*.

Results of the three sets of simulations are shown in Table 4, Table 5, and Table 6, respectively. In each table, the four columns under the title *Execution Result* show the states for the steps at lines 4, 5, 6, and 7, respectively, as being executed (passing or failing) or skipped, respectively. The last column indicates whether the steps are executed/skipped correctly (marked by "✓") or incorrectly (marked by "✗"). The correct behaviors are summarized below:

1. Checking of lift doors opening should be executed (line 7) only when the lift stops at the nearest floor (line 4);
2. Checking of lift doors opening should be skipped (line 7) if the lift fails to stop at a floor (line 4);
3. Regardless of the lift stopping or not (line 4), checking of warning (line 5) and checking of request canceling (line 6) are always executed.

Table 4 shows the simulation results in behave without turning off the failing-fast option. Due to sequential execution and failing-fast, the result is correct only when the lift behaves normally (Simulation 1). (There could be a short delay in warning and cancellation due to the gap in time between emergency occurrence and lift stopping. We assume the delay is short enough to be ignored.) If the warning step or the cancellation step fails, the check for the lift doors opening are incorrectly skipped due to the failing-fast feature (Simulations 2 and 3). Moreover, when the lift fails to stop at a floor, the checks for warning and cancellation are incorrectly skipped although the check for lift doors opening is correctly skipped (Simulation 4).

**Table 4.** Simulation results of executing the lift safety specification in Figure 3: behave with failing-fast.

| Simulation | Execution Result | | | | Result Correct? |
| :---: | :---: | :---: | :---: | :---: | :---: |
| | Lift Is Stopped (Line 4) | Emergency Indicator Is on (Line 5) | Request Is Canceled (Line 6) | Lift Doors Are Open (Line 7) | |
| 1 | executed-passing | executed-passing | executed-passing | executed | ✓ |
| 2 | executed-passing | executed-passing | executed-failing | skipped | ✗ |
| 3 | executed-passing | executed-failing | skipped | skipped | ✗ |
| 4 | executed-failing | skipped | skipped | skipped | ✗ |

Table 5 shows the simulation results in behave with the failing-fast option turned off. Since the option is applied at the scenario level (Section 2.2), all steps are executed regardless of the previous steps failing. When the lift stops at the nearest floor, the check for lift doors opening is still correctly executed even if the check for warning or the check for cancellation fails (Simulations 1–4). However, if the lift fails to stop at a floor, the check for lift doors opening is incorrectly executed, generating a misleading result (Simulations 5–8).

Table 6 shows the simulation results obtained by executing the specification of Figure 9 with concurrentSpec with keyword argument *continue_after_failure* = *true* for the warning step (line 5) and the cancellation step (line 6). If the lift stops at the nearest floor, the check for lift doors opening is correctly executed due to the keyword argument setting to continue after failure (Simulations 1–4). If the lift fails to stop, the check for lift doors opening is correctly skipped (Simulations 5–8) since the Then group fails and does not continue execution as a whole. Note that with the concurrency enhancement, the checks for warning and cancellation are executed simultaneously with the check for lift stopping. Thus, all checks are correctly made.

**Table 5.** Simulation results of executing the lift safety specification in Figure 3: `behave` without failing-fast.

| Simulation | Execution Result | | | | Result Correct? |
|:---:|:---:|:---:|:---:|:---:|:---:|
| | **Lift Is Stopped (Line 4)** | **Emergency Indicator Is on (Line 5)** | **Request Is Canceled (Line 6)** | **Lift Doors Are Open (Line 7)** | |
| 1 | executed-passing | executed-passing | executed-passing | executed | ✓ |
| 2 | executed-passing | executed-passing | executed-failing | executed | ✓ |
| 3 | executed-passing | executed-failing | executed-passing | executed | ✓ |
| 4 | executed-passing | executed-failing | executed-failing | executed | ✓ |
| 5 | executed-failing | executed-passing | executed-passing | executed | ✗ |
| 6 | executed-failing | executed-passing | executed-failing | executed | ✗ |
| 7 | executed-failing | executed-failing | executed-passing | executed | ✗ |
| 8 | executed-failing | executed-failing | executed-failing | executed | ✗ |

**Table 6.** Simulation results of executing the lift safety specification in Figure 9: `concurrentSpec` with the failing-fast option turned off at the right places.

| Simulation | Execution Result | | | | Result Correct? |
|:---:|:---:|:---:|:---:|:---:|:---:|
| | **Lift Is Stopped (Line 4)** | **Emergency Indicator Is on (Line 5)** | **Request Is Canceled (Line 6)** | **Lift Doors Are Open (Line 7)** | |
| 1 | executed-passing | executed-passing | executed-passing | executed | ✓ |
| 2 | executed-passing | executed-passing | executed-failing | executed | ✓ |
| 3 | executed-passing | executed-failing | executed-passing | executed | ✓ |
| 4 | executed-passing | executed-failing | executed-failing | executed | ✓ |
| 5 | executed-failing | executed-passing | executed-passing | skipped | ✓ |
| 6 | executed-failing | executed-passing | executed-failing | skipped | ✓ |
| 7 | executed-failing | executed-failing | executed-passing | skipped | ✓ |
| 8 | executed-failing | executed-failing | executed-failing | skipped | ✓ |

## 6. Consequences and Related Issues

The proposed concurrency enhancement of Gherkin and the tool `concurrentSpec` successfully resolve the specification errors in the two motivating examples. As with any solution, some consequences and related issues are incurred and merit further discussion.

### 6.1. Compatibility in Style

In the proposed enhancement, the keywords `And` and `But` are reserved for use as concurrent steps inside a sequential group that begins with keyword `Given`, `When`, or `Then`. Thus, when using `concurrentSpec`, it is necessary to represent two sequential steps as `Then-Then` rather than `Then-And` or `Then-But`, which can seem somewhat awkward. On the other hand, scenarios specifiable with the original Gherkin are specifiable with the proposed enhancement if all `And`'s and `But`'s are replaced by the closest `Given`, `When`, or `Then` keyword that appears before them. This is clearly a trade-off: at the sacrifice of a little awkwardness in writing style, the capability to specify concurrent behaviors is gained.

### 6.2. Collocating Specification and Continuation after Failure Settings

The proposed tool `concurrentSpec` is an embedded DSL. This has mixed consequences as we explain below using the lift safety specification as an example. On the negative side, the specification is embedded inside a Python program (Figure 9), which is less concise and lucid when compared with the specification written in a feature file (Figure 3). On the positive side, it is now feasible to collocate the behavior specification of a step with its continuation after failure setting. This ensures that the complete specification

is considered by the whole team when they read the specification written in the embedded DSL together. In contrast, with a tool such as behave, the behavior specification is written in a feature file while the continuation after failure setting is written as program code in a separate file. It is easy for the whole team to just concentrate on the feature file and leave settings of continuation after failure in the code to developers.

### 6.3. Implementation Level Concurrency

Let it be said out front that *concurrency is completely feasible with a tool such as* behave *if one is willing to blur the specification and leave concurrency up to implementation*; this is shown in Figure 10, a rewrite of the specification of scheduled sprinkling in Figure 2. Note, in particular, how all the concurrent steps are lumped into one step; see line 5 in Figure 10.

```
1  Scenario: scheduled sprinkling
2      Given three sprinklers A, B, and C
3        And the scheduled time has been set to 4:00:00 am
4       When the time is 4:00:00 am
5       Then sprinklers A, B, and C should emit water within 5 seconds
```

**Figure 10.** Scheduled sprinkling with behave: lumping three concurrent behaviors into one step.

Figure 11 shows the step definition function of the Then step initially generated with behave and later fixed by the developer. In the function, a Thread is used to run the assertion concurrently, one instance for each sprinkler.

However, the step definition of Figure 11 has the following limitations:

- Lumping the concurrent steps of the Then group of Figure 2 into one step hides the specific point of failure: when the step fails, it would only be possible to know that one of the sprinklers had failed but not which one had failed. This hampers the task of diagnosis;
- The level of abstraction is lowered from the specification level to the step definition at the implementation level. Specifically, unless the domain expert or analyst is comfortable working with code (and Thread in particular), he or she will have to depend on the developer or tester to translate the step at line 5 of Figure 10. In other words, the domain expert or analyst is deprived of his or her place to specify concurrent behaviors. Such delegated translation is exactly what BDD tries to eliminate as it creates gaps in communication [40];
- If the step fails, it is not directly possible for behave to report the error due to the use of Thread.

```
1  @then(u'sprinklers A, B, and C should emit water within 5 seconds')
2  def step_impl(context):
3      thread_a = Thread(target=check_sprinkler_a_is_emitting_water,
          args=[context])
4      thread_b = Thread(target=check_sprinkler_b_is_emitting_water,
          args=[context])
5      thread_c = Thread(target=check_sprinkler_c_is_emitting_water,
          args=[context])
6      # run threads and wait for threads to finish
```

**Figure 11.** Scheduled sprinkling with behave: implementing concurrency in a step definition.

## 7. Conclusions

Behavior-driven development (BDD) has brought the benefits of ubiquitous language, executable specification, and living documentation to mainstream software development. In order to bring these benefits to the development of the inherently concurrent Internet of Things (IoT) systems, it is necessary to have good support in the specification and execution of concurrent behaviors in IoT systems. Our contribution in this paper is to demonstrate that the current BDD specification language Gherkin and its associated tools are inadequate to the specification and execution of concurrent behaviors and proposed an enhancement to Gherkin and developed a supporting tool accordingly.

The enhancement to the Gherkin language introduces sequential groups of concurrent steps. Each group is led off with a step that begins with keyword `Given`, `When`, or `Then`. Groups are interpreted sequentially in the order they appear in a specification. Inside each group, the steps beginning with keyword `And` or `But` are interpreted as steps concurrent with the leading step. The enhancement only changes interpretation of the existing Gherkin keywords but does not introduce new keywords. Moreover, existing sequential Gherkin specification can be easily ported to use the enhancement by simply changing the keywords `And` and `But` in the steps to the closest keyword `Given`, `When`, or `Then` before them. We developed a tool called `concurrentSpec` in Python for correctly interpreting and executing the specification of concurrent behaviors written in the Gherkin enhancement. In addition, `concurrentSpec` also supports the continuation of execution after a step fails within a `Then` group. As demonstrated in the simulation results, the two motivating examples of concurrent behaviors were correctly resolved with the proposed Gherkin enhancement and `concurrentSpec`.

In our future work, we plan to explore the use of sequential groups of concurrent steps in the context of `Given` and `When` keywords by studying more IoT application scenarios from real world IoT systems or research literature. Under the parsimony principle, we will also explore new keywords, such as `Or` for even more expressiveness in specification. Lastly, we will add features to `concurrentSpec` such as feature organization, integration with user stories, and reporting.

**Author Contributions:** Conceptualization, B.-Y.W., Y.-C.Y. and Y.C.C.; software, B.-Y.W., Y.-C.Y. and Y.C.C.; writing—original draft preparation, B.-Y.W., Y.-C.Y. and Y.C.C.; writing—review and editing, B.-Y.W., Y.-C.Y. and Y.C.C.; supervision, Y.C.C.; funding acquisition, Y.C.C. All authors have read and agreed to the published version of the manuscript.

**Funding:** This research is supported by National Science and Technology Council, Taiwan.

**Institutional Review Board Statement:** Not applicable.

**Informed Consent Statement:** Not applicable.

**Data Availability Statement:** No new data were created or analyzed in this study. Data sharing is not applicable to this article.

**Conflicts of Interest:** The authors declare no conflict of interest.

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
