# Peer review of "Specifying Internet of Things Behaviors in Behavior-Driven Development: Concurrency Enhancement and Tool Support"

_applsci, doi:10.3390/app13020787_

Round 1
Reviewer 1 Report
Summary
The article investigates the application of the behavior-driven development method to IoT system projects. Specific attention is paid to concurrent behaviors and its handling in the context of BDD.
The article is clearly structured and well written.
Remarks
- Sources for figures should be provided - ("own picture" if applicable)
- It is not clear why concurrent behaviors should appear in IoT systems, only. Could there be any other applications, with concurrent behavior, too?
Reviewer 2 Report
I find the work is interesting, thank you very much. The flow of the work is easy to follow. The idea to enhance Gherkin with concurrent specification is interesting and novel. The simulation using time delay is a good way to simulate issues in concurrency and the results are easy to understand.
However, I have the following suggestion to make the paper more readable. I am not sure if it is a good idea to put algorithm under the figure label. I find the text the wording in the algorithm are rather small and difficult to read.
Section 6.4, I think the author is referring to the parallel processing capabilities in Gherkin tool which intends to speed up the test time. I find it has little things to do with this work. I suggest that the section to be removed or change the title of "parallelism vs. concurrency" to something more reflective to the content.
Overall it is a good work. Thank you.
